# Current Diagnostic and Therapeutic Approaches to Cytomegalovirus Infections in Ulcerative Colitis Patients Based on Clinical and Basic Research Data

**DOI:** 10.3390/ijms21072438

**Published:** 2020-03-31

**Authors:** Yoshihiro Yokoyama, Tsukasa Yamakawa, Takehiro Hirano, Tomoe Kazama, Daisuke Hirayama, Kohei Wagatsuma, Hiroshi Nakase

**Affiliations:** Department of Gastroenterology and Hepatology, Sapporo Medical University School of Medicine, Sapporo 060-8543, Japan; yoshi_yokoyamaa@yahoo.co.jp (Y.Y.); awakamay.dem1@gmail.com (T.Y.); a08m081@gmail.com (T.H.); t_kazama@kc5.so-net.ne.jp (T.K.); hirarin95@yahoo.co.jp (D.H.); waga_a05m@yahoo.co.jp (K.W.)

**Keywords:** human cytomegalovirus, HCMV colitis, inflammatory bowel disease, ulcerative colitis

## Abstract

Human cytomegalovirus (HCMV) is a ubiquitous herpesvirus (the human herpesvirus 5) and an opportunistic pathogen that primarily infects HIV-positive and other immuno-compromised patients. Retrospective studies in the field of inflammatory bowel disease (IBD) have suggested a relationship between a concomitant colonic HCMV infection and poor outcomes in patients with an ulcerative colitis (UC) due to the presence of HCMV in surgical specimens of patients with a toxic megacolon or a steroid-resistant UC. Therefore, gastroenterologists have focused on the contribution of HCMV infections in the exacerbation of UC. Numerous studies have addressed the benefits of treating colonic HCMV reactivation in UC using an antiviral treatment. However, its clinical relevance remains uncertain as only a few prospective studies have assessed the direct relationship between clinical outcomes and the viral load of HCMV in colonic tissues. HCMV reactivation can be triggered by inflammation according to fundamental research studies. Thus, optimal control of intestinal inflammation is essential for preventing an HCMV reactivation in the intestinal mucosa. Indeed, several reports have indicated the effectiveness of an anti-tumor necrosis factor-alpha (TNFα) treatment in patients with an active UC and concomitant HCMV infections. In this review, we describe the mechanism of HCMV reactivation in UC cases and discuss the current issues regarding diagnosis and treatment of HCMV infections in UC patients.

## 1. Introduction

Human cytomegalovirus (HCMV), a double-stranded DNA virus belonging to the family *Herpesviridae*, is generally contracted during childhood and can persist as a lifelong latent infection. Latent HCMV infection is reactivated under inflammatory conditions in an immunosuppressed host and can cause an organ damage [1]. Notably, 40%–100% of adults are carriers of HCMV with the seroprevalence typically increasing with age [2]. HCMV infections are diagnosed from clinical symptoms, organ lesions corresponding to the presented clinical symptoms, and evidence of virus in the diseased tissue through analysis of blood samples and other bodily fluids [3].

Ulcerative colitis (UC) is a non-specific chronic inflammatory bowel disease with ulcers and erosions in the colonic mucosa. It is thought to be caused by a multifactor involving genetic factors and environmental factors such as diet, but the etiology has not been clarified. UC mainly causes inflammation in the colonic mucosa, causing gastrointestinal symptoms such as diarrhea, hematochezia, and abdominal pain, often requiring immunosuppressive treatment.

The HCMV reactivation in healthy people, which is regulated by both HCMV-specific antibodies and CD8-positive T cells, is disrupted under the immunocompromised condition. Therefore, patients with medically refractory UC could be prone to the HCMV infection because of the use of immunosuppressive drugs, especially corticosteroid in addition to sustained inflammation in colonic mucosa triggering HCMV reactivation. The association between cytomegalovirus infections and UC was first recognized over 50 years ago [4]. The prevalence of HCMV reactivation in patients with severe UC was reported in 4.5%–16.6% of cases and approximately 25% in patients requiring colectomy for severe colitis [5]. HCMV infections in UC patients have been associated with steroid resistance and in severe cases, the infection has been associated with colectomy [5,6]. Clinically, the lack of established diagnostic methods and the concerns regarding the appropriate mechanism and the time of administration of an antiviral treatment presents an issue. In this review, we discuss the mechanisms of HCMV reactivation and the current issues related to the diagnosis and treatment of HCMV infections in UC patients.

## 2. Latent Infection and Reactivation of Human Cytomegalovirus (HCMV)

HCMV exhibits an extensive tissue affinity in the human body and is known to infect many cells and tissues such as macrophages, vascular endothelium, epithelial cells, fibroblasts, and nerve cells [7]. Conversely, previous clinical and fundamental studies have revealed that following a primary HCMV infection, the CD34+/CD33+ myeloid progenitor cells in the bone marrow and the CD14+ monocyte cells in the peripheral blood are latently infected. Furthermore, in vitro and ex vitro studies using these cells have shown that stimulation of the inflammatory cytokines, such as tumor necrosis factor-alpha (TNF-α), interferon-gamma (IFN-γ), and granulocyte-macrophage colony-stimulating factor (GM-CSF), causes an HCMV reactivation by differentiating the HCMV latent cells [8,9,10]. In immunocompetent hosts, HCMV reactivation is controlled by specific antibodies and the CD8-positive T cells (Figure 1).

Contrarily, this defense mechanism is disrupted in an immunosuppressed host and HCMV is reactivated, causing damage to multiple organs, including the gastrointestinal tract. Additionally, for an HMCV reactivation, it is imperative for the virus to express the immediate early (IE) genes that trigger viral growth, followed by transcription of the early (E) and the late (L) genes that result in the formation of viral proteins. Expression of the IE gene is controlled by the major immediate-early promoter (MIEP) located upstream, with various transcription factor-binding sites in the vicinity [11]. In particular, TNF-α, one of the important cytokines involved in the pathophysiology of inflammatory bowel disease (IBD), binds to the TNF receptor expressed on the cell surface and induces transcription of the IE gene through the activation of intracellular signals such as nuclear factor kappa B (NF-κB) [12]. Generally, inflammatory cytokines, including TNF-α, are important for the pathophysiology of IBD and the reactivation of HCMV in the intestinal tract by promoting the differentiation of HCMV latently-infected cells and the virus growth.

## 3. Methods for Diagnosis of HCMV Infections in Ulcerative Colitis (UC) Cases

It is difficult to distinguish clinically UC exacerbation from concomitant HCMV colitis because, typically, UC patients present with intestinal symptoms such as diarrhea and bloody stool. The various endoscopic findings are observed in UC complicated by HCMV infection. No definitive endoscopic findings lead to the diagnosis of concomitant HCMV infection in active UC. However, the existence of ulcerative lesions in UC patients refractory to immunosuppressive therapies may suggest the possibility of HCMV infection. Endoscopic findings such as punched-out ulcers, longitudinal ulcers, diffuse mucosal defect, and geographic ulcers (Figure 2) have been reported to be characteristics of UC complicated by HCMV infection [13,14].

However, it is clinically difficult to diagnose HCMV colitis only by endoscopic findings [15,16] and it remains necessary to prove the HCMV infection. There are several methods to diagnosis an HCMV infection. We should know the advantages and disadvantages of each diagnostic methods (Table 1).

### 3.1. HCMV Isolation in Culture

The most reliable test method in clinical virology is isolating HCMV in culture as it proves the existence of live HCMV with no false positives or negatives related to the detection of antigens and nucleic acids. Virus isolation in culture is performed using samples such as blood, tissue, and urine [17]. This test, however, requires 21 days to process the changes in cell culture and the sensitivity is also dependent on the isolation technique used [18]. The sensitivity and specificity of HCMV isolation in culture were reported in 45%–78% and 89%–100% of cases, respectively [19].

### 3.2. Diagnostic Serum Antibody Test for HCMV

Serological tests are used for diagnosis by determining the changes in IgM and IgG antibody titers over time by comparing the acute- and convalescent- phase paired sera (measured after 2 weeks-4 weeks). HCMV-IgM antibody titers increase from 2 weeks to 6 weeks after the initial infection, and usually decline within 2 months to 3 months in healthy individuals. IgM antibody is rarely detected in HCMV reactivation [20]. Recent HCMV infection is indicated by a significant, at minimum 4-fold rise in IgG antibody titer in the paired sera and the sensitivity and specificity were reported in 98%–100% and 95%–99% of cases, respectively [19]. However, a high IgG antibody titer persists throughout the life of HCMV-infected individuals, and the fluctuations in HCMV-IgG antibody titer due to virus reactivation is small. Thus, serum antibody tests are useful for screening HCMV-infected individuals that are vulnerable to an HCMV reactivation, but not necessarily for evaluating active infections, including an HCMV reactivation.

### 3.3. HCMV Antigenemia Assay

This method detects HCMV antigen-positive cells present in peripheral blood using a monoclonal antibody against the primary structural antigen (pp65) of HCMV. Neutrophils responsible for the phagocytosis of pp65 are detected, rather than the infected cells. Quantification of positive leukocytes allows for the degree of viremia to be quantified, which reflects both clinical symptoms and disease state. However, it must be recognized that peripheral blood reactivation does not necessarily reflect HCMV reactivation in the gastrointestinal tract. Additionally, HCMV antigenemia assays have high specificity and low sensitivity for moderate/severe UC cases that are complicated by the histopathological diagnosis of HCMV colitis [21,22]. Chun et al. [21] reported that the specificity and sensitivity of the HCMV antigenemia assay for diagnosis HCMV colitis were 87.1% and 66.7%, respectively. In addition, the specificity of the steroid-resistant UC cases was 91.7%, which was useful in making the decision of initiating antiviral treatment. Kim et al. [22] reported that the specificity and sensitivity of the HCMV antigen test were 81.7% and 47%, respectively.

### 3.4. HCMV-DNA Testing and Analysis

The HCMV-DNA test method is based on extracting a small amount of nucleic acid present in a specimen, followed by amplification and detection of DNA using DNA polymerase. Notably, the real-time polymerase chain reaction (PCR) method enables quantitative analysis and is used for monitoring HCMV infection after an organ or a bone marrow transplantation, and determining the therapeutic effect following treatment for HCMV infection. However, false positives due to DNA contamination and impedance of DNA amplification due to mutations in primer binding sites presents challenges. Blood HCMV-DNA testing is standard for monitoring HCMV infections in organ and bone marrow transplant patients, and its inclusion in the diagnosis algorithm in the UC-complicated HCMV infections has also been suggested [23]. This method is useful for diagnosing HCMV infections not only in blood but also in various organs, such as the colonic mucosa. We demonstrated the usefulness of the mucosal PCR in diagnosing HCMV infections in UC patients [24,25,26]. The sensitivity and specificity of the mucosal PCR tests were reported to be 65%–100% and 40%–92%, respectively. However, the high sensitivity of the quantitative real-time PCR assay may result in a low specificity for diagnosing active HCMV infections because HCMV-DNA in samples with low copy number may be detected by the PCR assay but may not actually reflect an active infection in the organs [6].

### 3.5. Histopathological Examination

The gold standard for diagnosing HCMV infections of each organ is the histological presence of an inclusion body. Typical HCMV infected cells are large and have cytomegalic inclusion bodies with a halo indicating an active HCMV proliferation. It is difficult to identify HCMV using only hematoxylin and eosin (HE) staining. Therefore, it is better to use immunohistochemistry (IHC) staining. The diagnostic method with histological examination affords high sensitivity (92% to 100%), but low and variable sensitivity (10% to 87%): a combination of HE and IHC staining increases the sensitivity to 78% to 93% [27]. The sensitivity of HCMV detection in tissues has been improved by using HCMV-specific antibodies and/or in situ DNA hybridization. Caution is required because the cytomegalovirus (CMV)-positive rates vary depending on the number of biopsy tests conducted and where the biopsy is performed [28]. As for optimal specimen number for HCMV diagnosis, one study recommended that a flexible sigmoidoscopy with 11 biopsies in UC and a colonoscopy with 16 biopsies in Crohn’s disease for assessing for HCMV infection [29]. However, such high number biopsies might be associated with risks of hemorrhage and perforation. Generally, HCMV-positive cells are more abundant at the base of ulcers than at the margin of ulcers [29]. However, Zidar et al. reported that no marked difference between the ulcer base and edge was evident in terms of the highest densities of HCMV-positive cells [30]. Taken together, we recommend that it could be better to take biopsy specimens from the edge of ulcers in cases of deep ulceration that active UC patients suffer for the investigation of HCMV infection.

## 4. Treatment of UC Patients with Concomitant HCMV Infection

### 4.1. When to Start Antiviral Treatment?

The treatment of HCMV infection in patients with active UC is complicated by the difficulty experienced in exactly distinguishing an HCMV reactivation from HCMV colitis, as inflammatory conditions in the colonic mucosa of UC patients may contribute to an HCMV reactivation [23,31]. When treating UC patients with resistance for anti-inflammatory treatment, we need to perform examination for diagnosis of HCMV infections. Based on each of the results, we make the decision whether to start antiviral treatment (Figure 3).

#### 4.1.1. HCMV Antigenemia Assay

The HCMV antigenemia assay is widely used for decision of initiating antiviral treatment. However, the use of HCMV antigen tests as diagnostic tools for HCMV colitis remains unclear. In clinical practice, we often encounter that the HCMV antigen is not detected even if HCMV infection in patients with active UC is suspected. Matsuoka et al. [32] reported that in patients undergoing UC treatment, the HCMV antigenemia assay may be negative by reducing the dose of prednisolone, and they concluded that detection of HCMV antigen does not necessarily indicate the requirement of antiviral treatment. This result suggested that the result of the HCMV antigenemia assay may depend on disease activity and the dose of immunosuppressants such as steroids. In fact, no cut-off value of the HCMV antigenemia assay for diagnosis of HCMV colitis has yet been established.

#### 4.1.2. Histopathological Examination

As mentioned above, the histological test confirming the presence of HCMV using HE or IHC staining is the gold standard for diagnosis. Therefore, if the histological examination shows the existence of HCMV in colonic mucosa, antiviral treatment should be considered, although it may depend on the number of HCMV-positive cells.

#### 4.1.3. HCMV-DNA Testing and Analysis

In HCMV-DNA analysis, the initiation of antiviral treatment is considered for patients with a high viral load in colonic mucosa [5,28]. However, a clear cut-off value of HCMV-DNA in colonic mucosa regarding the initiating antiviral treatment does not exist. Some studies reported the cut-off value for mucosal viral load of IBD patients. Roblin et al. [33] reported that administration of antiviral drugs led to a remission in 7 out of the 8 steroid-resistant UC patients with 250 copies/mg or higher of HCMV-DNA in intestinal tissues. Ciccocioppo et al. [34] compared mucosal HCMV-DNA value between IBD patients (refractory and non-refractory) and controls. The refractory IBD patients showed DNA peak values of more than 10^3^ copies/10^5^ cells in diseased mucosa in comparison to non-diseased mucosa, while non-refractory patients and control displayed levels below this threshold, thus allowing HCMV colitis to be distinguished from mucosal infection. Also, Okahara et al. [35] suggested that the low viral load of HCMV in colonic mucosa might not contribute to exacerbation of UC. In case with low viral load of HCMV, it is necessary to refer to other HCMV tests and endoscopic findings, and antiviral treatment should be considered depending on the severity of UC. Lawlor et al. [31] reported that the HCMV antigenemia assay and mucosal HCMV-DNA test have false positives and recommend using HCMV-DNA tests in the peripheral blood to test for an HCMV infection.

### 4.2. Efficacy of the Antiviral Treatment

The effects of antiviral treatment on UC with HCMV colitis have been observed in a few studies. Recently, in a meta-analysis of 15 studies, stratified analysis focused on cases of steroid resistance showed that the colectomy rate was significantly lower in the group treated with antiviral treatment than in the control group [36]. Jones et al. [28] examined the biopsy tissue with IHC staining and examined the prognosis retrospectively for the groups with both high- and low-density inclusion bodies. In the high-density group, antiviral treatment significantly reduced the surgical rate. Regarding colectomy, Zagórowicz et al. [37] reported that the group with numerous HCMV-infected cells in their intestinal tissue showed a significant decrease in colectomy rate with antiviral treatment. Kim et al. [38] conducted a prospective study of 72 UC patients who were moderately or severely treated with steroids. Ganciclovir was given to 14 patients who were ineffective with steroids and 11 patients (79%) of them showed improvement. Yilmaz et al. [39] reported a meta-analysis regarding the efficacy of ganciclovir among UC patients detected with HCMV infection. The result showed that administration of ganciclovir did not contribute to surgical resection and death related to UC. Taken together, these data reflect the effect of antiviral treatment on induction of clinical remission and reduction of surgical resection rate in a subpopulation of active UC patients concomitant with HCMV infection, and the difference of HCMV diagnostic methods regarding the initiation of antiviral treatment. Currently, few prospective observational studies are available and, therefore, further studies will be required to make decisions on antiviral treatment.

### 4.3. Anti-Inflammatory Treatment for Ulcerative Colitis

In UC cases that are complicated with HCMV infection, suppression of HCMV reactivation by anti-inflammatory treatment is also important. The risk of complications accompanied by HCMV infection is high when steroids and cyclosporine are administered due to strong suppression of the immune system of the host [40,41,42]. European Crohn’s and Colitis Organisation guideline state that not all patients need to be screened for HCMV infection before starting immunosuppression therapy, but in steroid-resistant colitis, HCMV should be examined prior to increasing treatment [43]. It is important to reduce the amount of steroids promptly to decrease the risk of infection by HCMV [38]. Hissong et al. [44] reported that HCMV colitis rates are decreasing among patients with IBD, reflecting the change from steroids-based treatment to more effective agents. This data suggests that withdrawing from steroidal dependence could reduce the risk of HCMV colitis.

Additionally, we have reported on the effectiveness of granulocyte/monocyte adsorptive apheresis (GMAA) for UC-concomitant HCMV colitis. It appears that the GMAA has little effect on HCMV reactivation, because the GMAA does not directly affect the local immune system. It may be considered as an induction treatment to UC patients with a history of an HCMV infection [45,46]. Considering the role of TNF-α in HCMV reactivation, anti-TNFα treatment could be one of the promising treatment options for UC-concomitant HCMV infection, especially in cases with high HCMV viral load. In fact, studies have shown that anti-TNFα treatment does not influence an increased risk of HCMV reactivation [47,48,49].

In recent years, the evolution of biological products has been remarkable, and new treatments suppressing the HCMV reactivation will appear.

## 5. Experimental Inflammatory Bowel Disease (IBD) Mouse Model with Concomitant Cytomegalovirus (CMV) Infection

It has been hypothesized that IBD cases concomitant with CMV infections have a poor prognosis. However, the exact mechanism by which HCMV exacerbates IBD remains unclear. We had previously reported an experimental IBD mouse model that was exacerbated by a CMV infection [50,51]. The mouse cytomegalovirus (MCMV) shares a high sequence homology with HCMV and serves as a useful tool for understanding the pathogenesis of HCMV. T-cell receptor (TCR) -α knockout (KO) mice develop spontaneous colitis, similar to the immune response of UC. We established MCMV latently-infected TCR-αKO mice that mimicked an HCMV latently-transmitted infection. This study demonstrated that IHC staining showed an increase in the MCMV-infected cells as colitis occurred in the TCR-αKO mice. Cells infected with MCMV were also detected primarily in the inflamed colonic mucosa, which was compatible with clinical data indicating that HCMV was present at sites of inflammation rather than at non-inflammatory sites [52]. HCMV also induces neutrophil migration and is known to reprogram monocyte differentiation into M1 macrophages in vitro [53]. In fact, we found more migrating neutrophils and M1 macrophages at the site of inflammation of the colon in this mouse model. IHC staining showed that MCMV potentially infected the perivascular stromal cells, including pericytes. These data indicate that HCMV infection in the colonic mucosa of UC patients may spread from the perivascular stromal cells to the endothelial and epithelial cells as colitis progresses. The application of this model may aid in elucidating a detailed mechanism of IBD deterioration triggered by an HCMV infection.

## 6. Conclusions

Reactivation of HCMV causes exacerbation of disease activity of UC patients. The diagnosis continues to be based on HCMV detection in histopathology, but we believe that the quantitative PCR method is a promising alternative approach to better define the extent of an HCMV reactivation in histological negative cases. The indications for antiviral therapy in IBD remain controversial, but more recent findings suggest the benefit of antiviral treatment in patients with a histologically proven HCMV colitis or a high HCMV-DNA load in colonic tissues determined by quantitative PCR. In addition, insufficient control of inflammation can reactivate HCMV, resulting in the increasing disease activity of the UC. Therefore, optimal control of colonic inflammation must be achieved in UC patients positive for HCMV infections who are resistant to conventional immunosuppressive treatment. Further experimental data and prospective studies will be required in the future to develop appropriate therapeutic approaches for active UC patients complicated with HCMV infection.

## Figures and Tables

**Figure 1 ijms-21-02438-f001:**
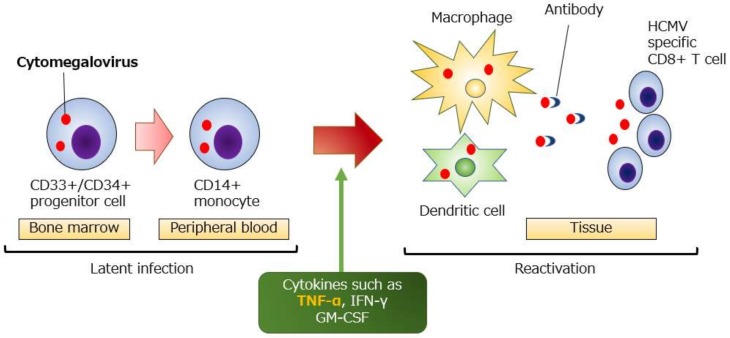
Latent infection and reactivation of human cytomegalovirus (HCMV). HCMV latently infects the CD33+/CD34+ cells in the bone marrow and the differentiated CD14+ monocytes in the peripheral blood. HCMV reactivation occurs when HCMV-infected cells enter the tissue and differentiate into macrophages or dendritic cells. Locally-reactivated HCMV infection is prevented from spreading to the whole body by anti -HCMV antibodies and CD8+ T cells. TNF-α: tumor necrosis factor-alpha, IFN-γ: interferon-gamma, GM-CSF: granulocyte-macrophage colony-stimulating factor.

**Figure 2 ijms-21-02438-f002:**
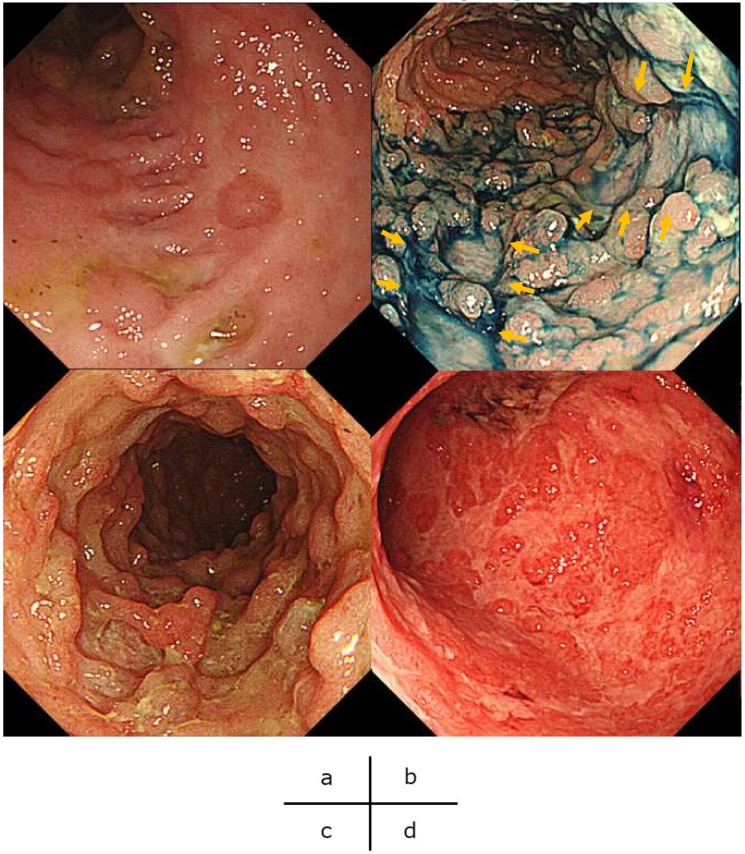
Characteristic endoscopic findings of ulcerative colitis complicated by human cytomegalovirus infection. (**a**) Punched-out ulcers: the round shape of ulcerations with clear demarcation. (**b**) Longitudinal ulcers: ulcerations running along the lumen of the colon (arrows). (**c**) Diffuse mucosal defect: many mucosal defects with surrounding reddish and erythematous mucosa. (**d**) Geographic ulcers: shallow and widespread mucosal defects with unclear demarcation.

**Figure 3 ijms-21-02438-f003:**
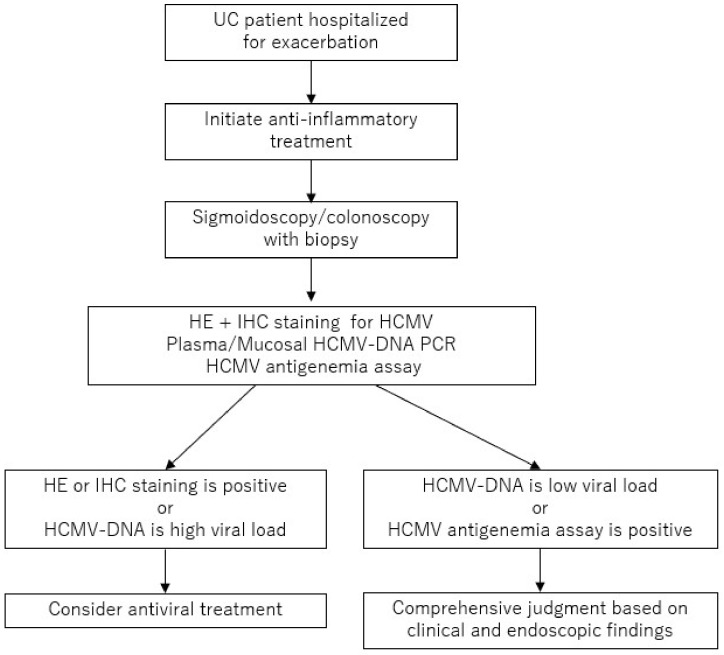
Proposed diagnostic and therapeutic strategy for the patients with ulcerative colitis and the patients with a suspected human cytomegalovirus colitis. UC: ulcerative colitis, HE: hematoxylin and eosin, IHC: immunohistochemistry, HCMV: human cytomegalovirus, PCR: polymerase chain reaction.

**Table 1 ijms-21-02438-t001:** Diagnostic tests for detecting HCMV infection. Each test has its own advantages and disadvantages.

Methods	Biological Specimens	Techniques	Advantages	Disadvantages
Isolation culture [17,18,19]	Blood, Biopsy, Urine	Culture	High specificity.	Long incubation time. Technical error.
Serum HCMV antibody [19,20]	Blood	ELISA	Marker of previous infection.	Not reflect reactivation.
HCMV antigenemia assay [21,22]	Blood	pp65 antigenemia	Presence of active blood infection. High specificity.	Low sensitivity in colitis.
HCMV-DNA [6,23,24,25,26]	Blood, Biopsy	PCR	High sensitivity.	Low specificity (difficult to differentiate between infection and disease). Cut-off not defined. No standardized technique.
Histopathology [27,28,29,30]	Biopsy	HE staining IHC staining	High specificity.	Sampling error.

ELISA: enzyme-linked immunosorbent assay, PCR: polymerase chain reaction; HE: hematoxylin and eosin, IHC: immunohistochemistry.

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
