# Peer review of "Current Diagnostic and Therapeutic Approaches to Cytomegalovirus Infections in Ulcerative Colitis Patients Based on Clinical and Basic Research Data"

_ijms, 2020, doi:10.3390/ijms21072438_

Round 1

Reviewer 1 Report

The present review paper “Current diagnostic and therapeutic approaches for cytomegalovirus infection in ulcerative colitis patients based on clinical and basic research data” submitted by Yoshihiro et al has described the role of HCMV in ulcerative colitis (UC). There are increasing cases of HCMV infection and UC cases. The authors have described the latent HCMV infection, description of HCMV diagnosis in UC cases using various assays, strategy for HCMV drug therapy, and an animal model. The paper is well written and covers most of the aspects of HCMV and UC. The current paper advances our knowledge of HCMV roles in ulcerative colitis. The current paper can be improved by addressing the few comments listed below.

Minor comments

1. The authors are requested to provide more details of figure 2 and explain elaborately.

2. The authors are suggested to include references in table 1 for each diagnostic test.

3. It is not essential but will improve the quality of paper if authors can include a paragraph for HCMV vaccine development and UC.

4. The authors are also suggested to provide epidemiological data on HCMV and UC if it is available.

5. The authors are also suggested to include if there is any role of HCMV in UC related cancer development.

6. The authors are requested to provide a substantial analysis of UC and HCMV infection in healthy and immunocompromised patients.

7. Healthy individuals do not show any symptoms of HCMV infection. The authors are requested to provide details on UC in non-CMV infected patients. The symptoms arise in vulnerable populations after immunosuppression by drugs that reactivate CMV in these patients. The authors should elaborate and explains it.

Author Response

Reviewer 1:

Comments and Suggestions for Authors

The present review paper “Current diagnostic and therapeutic approaches for cytomegalovirus infection in ulcerative colitis patients based on clinical and basic research data” submitted by Yoshihiro et al has described the role of HCMV in ulcerative colitis (UC). There are increasing cases of HCMV infection and UC cases. The authors have described the latent HCMV infection, description of HCMV diagnosis in UC cases using various assays, strategy for HCMV drug therapy, and an animal model. The paper is well written and covers most of the aspects of HCMV and UC. The current paper advances our knowledge of HCMV roles in ulcerative colitis. The current paper can be improved by addressing the few comments listed below.

Minor comments

  1. The authors are requested to provide more details of figure 2 and explain elaborately.

Thank you for your valuable comment. As you suggested, we changed the sentence in line 92 of page3 and the legends of Figure 2 in line 101 of page 4 as follows: “The various endoscopic findings are observed in UC complicated by HCMV infection. No definitive endoscopic findings were leading to the diagnosis of concomitant HCMV infection in active UC. However, the existence of ulcerative lesions in UC patients refractory to immunosuppressive therapies may suggest the possibility of HCMV infection.”, “(a) Punched-out ulcers: the round shape of ulcerations with clear demarcation. (b) Longitudinal ulcers: ulcerations running along the lumen of the colon (arrows). (c) Diffuse mucosal defect: many mucosal defects with surrounding reddish and erythematous mucosa. (d) Geographic ulcers: shallow and widespread mucosal defects with unclear demarcation.”, respectively. Also, we added arrows at the unclear lesions in the Figure 2-b.

  1. The authors are suggested to include references in table 1 for each diagnostic test.

Thank you for your comment. According to your comment, we include references each diagnostic test in Table 1 as follows:” Isolation culture [17-19], Serum HCMV antibody [19, 20], HCMV antigenemia assay [21,22], HCMV-DNA [6,23-26], Histopathology [27-30], respectively.

  1. It is not essential but will improve the quality of paper if authors can include a paragraph for HCMV vaccine development and UC.

Thank you for your valuable comment. As you pointed out, HCMV vaccine is important for public health and preventive medicine, especially for congenital cytomegalovirus infections. However, since it is not put to practical use at this moment and this review mainly deals with the reactivation of HCMV, it will be omitted for HCMV vaccine in this review.

  1. The authors are also suggested to provide epidemiological data on HCMV and UC if it is available.

Thank you for your comment. According to your comment, we added the following sentence to the Introduction section in line 52 of page 2 as follow:” The prevalence of HCMV reactivation in patients with severe UC was reported 4.5-16.6% and approximately 25% in patients requiring colectomy for severe colitis [5].”

  1. The authors are also suggested to include if there is any role of HCMV in UC related cancer development.

Thank you for your great comment. The contribution of HCMV to UC related colorectal cancer has not been elucidated. It was reported that MCMV latently-infected TCR-αKO mice had the upregulated gene expression of proinflammatory cytokines in colonic mucosa including IL-6, IL-17, etc. Based on this experimental data, sustained inflammation triggered by HCMV infection could lead to the onset or development of colitis-associated cancer. However, further research will be required to clarify the association between HCMV and UC related colorectal cancer.

  1. The authors are requested to provide a substantial analysis of UC and HCMV infection in healthy and immunocompromised patients.

Thank you for your comment. According to your comment, we added some description to the Introduction section in line 43 of page 2 as follows:” The HCMV reactivation in healthy people, which is regulated by both HCMV-specific antibodies and CD8-positive T cells, is disrupted under the immunocompromised condition. Therefore, Patients with medically refractory UC could be prone to the HCMV infection because of the immunosuppressive drugs use, especially corticosteroid in addition to sustained inflammation in colonic mucosa triggering HCMV reactivation.”

  1. Healthy individuals do not show any symptoms of HCMV infection. The authors are requested to provide details on UC in non-CMV infected patients. The symptoms arise in vulnerable populations after immunosuppression by drugs that reactivate CMV in these patients. The authors should elaborate and explains it.

Thank you for your great comments. As you commented, we added the following sentences regarding UC in non-HCMV infected patients to the Introduction section in line 38 of page 1. “Ulcerative colitis (UC) is a nonspecific chronic inflammatory bowel disease with ulcers and erosions in the colonic mucosa. It is thought to be caused by a multifactorial disease involving genetic factors and environmental factors such as diet, but the etiology has not been clarified. UC mainly causes inflammation in the colonic mucosa, causing gastrointestinal symptoms such as diarrhea, hematochezia, and abdominal pain, often requiring immunosuppressive treatment.”. Explanation of HCMV reactivation in UC patients by immunosuppressive drug and inflammation in colonic mucosa was changed as mentioned above.

Reviewer 2 Report

It is well written review article for therapeutic approaches for cytomegalovirus infections in ulcerative colitis patients.

Minor points.

1) Page 4, 3.1. and 3.2.; Is there any data that shows sensitivity and specificity of HCMV isolation in culture and serum antibody test for HCMV?

2) Page 5, 4. Treatment of UC patients with concomitant HCMV infection; In UC patients with HCMV, Is there any data about the frequency of patients who have undergone sepsis or toxic megacolon requiring surgery, such as total colectomy?

Author Response

Reviewer 2:

Comments and Suggestions for Authors

It is well written review article for therapeutic approaches for cytomegalovirus infections in ulcerative colitis patients.

Minor points.

  • Page 4, 3.1. and 3.2.; Is there any data that shows sensitivity and specificity of HCMV isolation in culture and serum antibody test for HCMV?

Thank you for your valuable comment. Römkens et al reported a systematic review about diagnostic methods of HCMV infection in IBD patients. In this review, the sensitivity and specificity of HCMV isolation in culture were 45-78% and 89-100%, the sensitivity and specificity of serum antibody test for HCMV were 98-100% and 95-99%, respectively. As you commented, we should note the sensitivity and specificity of HCMV isolation in culture and serum antibody test for HCMV. We added some description in line 117 of page 4, in line 125 of page 4 as follows: “The sensitivity and specificity of HCMV isolation in culture were reported 45-78% and 89-100%, respectively [19].”, “and the sensitivity and specificity were reported 98-100% and 95-99%, respectively [19].”, respectively. And we added the systematic review by Römkens et al to the reference [19] in line 335 of page 9.

2) Page 5, 4. Treatment of UC patients with concomitant HCMV infection; In UC patients with HCMV, Is there any data about the frequency of patients who have undergone sepsis or toxic megacolon requiring surgery, such as total colectomy?

Thank you for your comment. Sager et al [5] reported that approximately 25% UC patients concomitant HCMV required colectomy for severe colitis. We described it to the Introduction section in line 52 of page 2 as follows. “The prevalence of HCMV reactivation in patients with severe UC was reported 4.5-16.6% and approximately 25% in patients requiring colectomy for severe colitis [5].”